# The Blood Microbiome and Health: Current Evidence, Controversies, and Challenges

**DOI:** 10.3390/ijms24065633

**Published:** 2023-03-15

**Authors:** Hong Sheng Cheng, Sin Pei Tan, David Meng Kit Wong, Wei Ling Yolanda Koo, Sunny Hei Wong, Nguan Soon Tan

**Affiliations:** 1Lee Kong Chian School of Medicine, Nanyang Technological University Singapore, Singapore 308232, Singapore; sunny.wong@ntu.edu.sg (S.H.W.); nstan@ntu.edu.sg (N.S.T.); 2Radiotherapy and Oncology Department, Hospital Sultan Ismail, Jalan Mutiara Emas Utama, Taman Mount Austin, Johor Bahru 81100, Malaysia; 3School of Biological Sciences, Nanyang Technological University Singapore, Singapore 637551, Singapore

**Keywords:** host–microbe interaction, microbial commensalism, dysbiosis, bacterial translocation, septicaemia

## Abstract

Blood is conventionally thought to be sterile. However, emerging evidence on the blood microbiome has started to challenge this notion. Recent reports have revealed the presence of genetic materials of microbes or pathogens in the blood circulation, leading to the conceptualization of a blood microbiome that is vital for physical wellbeing. Dysbiosis of the blood microbial profile has been implicated in a wide range of health conditions. Our review aims to consolidate recent findings about the blood microbiome in human health and to highlight the existing controversies, prospects, and challenges around this topic. Current evidence does not seem to support the presence of a core healthy blood microbiome. Common microbial taxa have been identified in some diseases, for instance, *Legionella* and *Devosia* in kidney impairment, *Bacteroides* in cirrhosis, *Escherichia/Shigella* and *Staphylococcus* in inflammatory diseases, and *Janthinobacterium* in mood disorders. While the presence of culturable blood microbes remains debatable, their genetic materials in the blood could potentially be exploited to improve precision medicine for cancers, pregnancy-related complications, and asthma by augmenting patient stratification. Key controversies in blood microbiome research are the susceptibility of low-biomass samples to exogenous contamination and undetermined microbial viability from NGS-based microbial profiling, however, ongoing initiatives are attempting to mitigate these issues. We also envisage future blood microbiome research to adopt more robust and standardized approaches, to delve into the origins of these multibiome genetic materials and to focus on host–microbe interactions through the elaboration of causative and mechanistic relationships with the aid of more accurate and powerful analytical tools.

## 1. Introduction

Blood is commonly thought to be sterile, except for in certain medical conditions such as septicaemia, whereby systemic infections of bacteria, viruses and fungi occur. The gold standard to detect living microbes in the bloodstream is blood culture. However, the emergence of next generation sequencing (NGS) technology has given rise to highly sensitive approaches such as 16S sequencing and shotgun metagenomics for microbial detection and taxonomic classification in an untargeted manner [1,2]. These NGS-based approaches are widely used to profile microbial populations in different compartments, including the gut [3], airway [4], skin [5], oral cavity [6] and urogenital tract [7], markedly advancing our understanding of the human microbiome and host–microbe relationship in physiological and various pathological contexts. Interestingly, many microbial analyses of blood specimens have detected bacteria and their genetic materials even in healthy individuals [8,9], leading to the conceptualization of a human blood microbiome and bringing the dogma of blood sterility into question.

The blood microbiome is defined by the assemblage of living microorganisms present in the blood. In human microbiome research, identifying a universal group of microbes (i.e., core microbiome) at a specific anatomical site that is shared by most humans and is crucial for host biological function is a major goal [10]. Likewise, the presence of a core healthy blood microbiome is postulated, the disturbance of which could potentially contribute to various diseases. Thus far, blood microbiome profiles have been assessed not only in healthy individuals, but also in patients with various health complications. The data reveal exciting clinical prospects of using blood microbiome genetic signature for risk stratification, diagnosis, disease surveillance and drug development (Figure 1). Nonetheless, the existence of the human blood microbiome remains highly debatable. The controversies typically revolve around two issues, namely the high risk of microbial contamination in low-biomass samples and the undetermined viability of blood microbiota based on culture-independent profiling methods. Furthermore, there is also no consensus on the research methodology and experimental controls to enable contaminant-free blood microbiome studies. As the concept of the blood microbiome has been well-summarized [8,9,11], our review aims to consolidate recent findings about the blood microbiome and to highlight the controversies, prospects, and challenges of the research topic.

## 2. Characterization of a Healthy Core Blood Microbiome

Emerging evidence highlights the presence of microorganisms in the blood of healthy individuals. For instance, bacterial growth was detected using aerobic and anaerobic blood cultures in up to 60% of donated blood packs [12]. Using powerful PCR and NGS techniques, 16S rRNA was found in 100% of blood samples [13,14]. Microbial DNA has also been reported in the blood from neonates [15] or even other animal species, such as dogs [16]. In contrast, some studies concluded that the presence of bacteria in blood is an unusual event because most healthy donors (78% to 84%) do not carry any bacterial species in their blood [17,18]. The large variation in the frequency of blood microbe-positive cases could be attributable to different experimental controls and decontamination strategies. In a recent preprint by Tan et al. (2022) [17], which aimed to investigate the presence (or absence) of a healthy core blood microbiome, the team performed shotgun metagenomic profiling of one of the largest multicentre cohorts from the SG10K_Health dataset (*n* = 9770 healthy donors), followed by a set of stringent quality control criteria and decontamination filters to minimize artefacts from sequencing, host DNA, taxonomic assignment, and batch-specific contaminants during sample preparation and from the reagent kits. They discovered that microbial DNA was present only in 16% of the healthy individuals at a median of one microbial species per individual [17]. In view of the ubiquity of environmental contamination at every step of sample processing, such stringent filters are effective in improving the signal-to-noise ratio, albeit at the expense of masking some genuine signals.

Additionally, there is no consensus on the composition of a healthy blood microbiome based on existing studies. *Staphylococcus* spp. is a common genus found in blood [12,14,17,18], yet information at the species level remains scarce and poorly characterized. Other potential blood microbial signatures include the *Proteobacteria* phylum [14] and *Cutibacterium acnes* [17]. There is remarkable bacterial diversity between different blood fractions: buffy coat, red blood cells and plasma [12,14]. The analysis using the SG10K_Health dataset revealed 117 blood microbial species comprising 110 bacteria, 5 viruses and 2 fungi [17]. Most of the species are commensals from the gut, mouth and urogenital tract, however, none of them demonstrated a clear cooccurrence trend nor were consistently detected in more than 5% of the donors, underscoring a significant intersubject variability that was also seen in neonatal blood [15,17]. Thus far, the varying results in the prevalence of blood microbes and the bacterial composition do not support the hypothesis of a core healthy blood microbiome for host functionality. Findings of the large SG10K healthy cohort strongly suggest a transient and sporadic translocation of commensals into the bloodstream which are quickly cleared out and do not lead to prolonged colonization [17]. On the other hand, the persistence of blood microbes could signify the onset of certain diseases, which will be elaborated below. While further validation from other human cohorts is crucial, the initiative to employ a stringent analytical pipeline and decontamination filters is commendable and could potentially be adopted in future blood microbiome research.

## 3. Clinical Significance of the Blood Microbiome in Human Diseases

Although the concept of a shared healthy blood microbiome remains inconclusive, disruption of mucosal integrity in certain disease states may exacerbate microbial translocation, leading to the persistence of microbes in the bloodstream. Unsurprisingly, many studies have profiled the dynamic changes in the blood microbiome during various pathological conditions, including cardiometabolic diseases, malignancies, inflammatory and immune disorders, and hepatic, respiratory and dermal diseases. Next, we will summarize recent evidence of the blood microbiome in human diseases.

### 3.1. Cardiometabolic Diseases

Linkages between blood microbial signatures and cardiovascular events and chronic metabolic diseases have been reported (Table 1). Three case-control studies revealed reduced blood microbial diversity in patients with myocardial infarction (MI) and chronic coronary syndrome compared to healthy individuals [19,20,21]. Similarly, a longitudinal cohort study concluded that Eubacteria was inversely correlated with the onset of cardiovascular complications [22]. It is worth noting that the study had disproportionate case (*n* = 73) and control (*n* = 3963) arms and only Eubacteria and phylum Proteobacteria levels at baseline were used for analysis. Nonetheless, numerous bacterial taxa were found to be associated with cardiovascular events across different studies. In the Oslo II patient cohort, the genera *Kocuria* and *Enhydrobacter* were positively associated with cardiovascular mortality, while the genus *Paracoccus* reduced the risk [23]. Conversely, Amar et al. (2019) reported the genus *Hymenobacter* as a positive predictor and four genera (i.e., *Brevundimonas*, *Chryseobacterium*, *Gordonia* and *Microbacterium*) of MI [19]. In another smaller case-control study, a higher abundance of the genus *Bifidobacterium* and a reduced abundance of the phylum Bacteroidetes were found in the blood of MI patients [20]. Remarkable microbial differences are also noted in individuals with acute and chronic coronary syndrome, whereby the former has more phyla Proteobacteria and Acidobacteriota, while the latter has more phylum Firmicutes and genus *Lactobacillus* [21]. Current evidence is overly intricate to pinpoint specific MI-driving blood microbes. Furthermore, the lack of common microbial taxa associated with cardiovascular events across different studies also highlights a diverse blood microbiome in MI patients, which could be attributed to known modifiers of the human microbiota, such as geographical regions, dietary patterns, host genetic, and environmental factors [24]. Interestingly, bacterial families and genera associated with cholesterol and lipid metabolism were markedly suppressed in MI patients, which may promote plaque formation in the coronary arteries [19,20]. Hence, functional analysis is a new strategy that may enable the reconciliation of discrepant microbial targets associated with cardiovascular events from various studies.

Apart from cardiovascular events, some studies have investigated chronic metabolic diseases such as hypertension and type 2 diabetes mellitus (T2DM). D.E.S.I.R. cohort revealed a higher 16S rDNA concentration in those who eventually developed T2DM and abdominal adiposity [25]. A prospective cohort study with primarily Chinese volunteers uncovered the predictive value of the blood genus *Bacteroides* for T2DM [26] and of the genera *Acinetobacter*, *Sphingomonas* and *Staphylococcus* for hypertension [27]. Despite these exciting results, more research is warranted to validate the causative relationship of the target bacteria with the associated diseases.

**Table 1 ijms-24-05633-t001:** Summary of investigations on the blood microbiome in cardiovascular disease and chronic metabolic diseases.

No.	Disease	Study Design	Sample Size	Detection Method	Findings	Reference
1.	Myocardial infarction	Case-control study	Case = 103; Control = 99	16S rDNA (V3-V4 region) sequencing	16S rDNA concentration was higher in MI patients but with a lower α-diversity. Caulobacterales order and Caulobacteraceae family were reduced in MI group. Cholesterol-degrading bacteria (Norcardiaceae, Aerococcaceae, *Gordonia*, *Propionibacterium*, *Chryseobacterium*, *Rhodococcus*) were decreased in MI patients.	[19]
2.	Myocardial infarction	Case-control study	Case = 29; Control = 29	16S rDNA (V3-V4 region) sequencing	Reduced α-diversity in MI group. Phylum Actinobacteria, class Actinobacteria, order Bifdobacteriales, family Bifidobacteriaceae, and genus *Bifidobacterium* were more abundant in MI group. Phylum Bacteroidetes, class Bacteroidia, and order Bacteroidales were reduced in MI group.	[20]
3.	Acute (ACS) and chronic coronary syndrome (CCS)	Case-control study	Case (ACS) = 70; Case (CCS) = 70; Control = 70	16S rDNA (V3-V4 region) sequencing	ACS and CCS had the highest and lowest α-diversity, respectively. Blood microbial communities of the three groups formed distinct clusters in Bray-Custis-based PCoA. Enriched taxa in healthy group: phylum Actinobacteria & Genus Staphylococcus; in CCS: phylum Firmicutes and genus *Lactobacillus*; in ACS: phyla Proteobacteria and Acidobacteriota.	[21]
4.	Cardiovascular disease (CVD)	Prospective cohort study (D.E.S.I.R.)	CVD = 73; No CVD = 3863	qPCR of conserved 16S rDNA regions	Phylum Proteobacteria was positively correlated with Eubacteria. Eubacteria was inversely correlated with cardiovascular events. Proteobacteria enhanced cardiovascular risk.	[22]
5.	Cardiovascular mortality	Case-control study (Oslo II)	Case = 227; Control = 178	16S rDNA (V3-V5 region) sequencing	78 genera from 6 phyla were found in the blood samples. Genera *Kocuria* and *Enhydrobacter* were associated with higher CVD mortality. Genus *Paracoccus* was inversely related with the outcome.	[23]
6.	T2DM & obesity	Prospective cohort study (D.E.S.I.R.)	T2DM = 131; No T2DM = 3149	qPCR of conserved 16S rDNA regions	16S rDNA concentration was higher in those who later developed T2DM and abdominal obesity. Healthy and T2DM individuals shared a similar blood microbiome, largely comprised of phylum Proteobacteria ( >85%).	[25]
7.	T2DM	Prospective cohort study (135)	T2DM = 50; No T2DM = 100	16S rDNA (V5-V6 region) sequencing	No difference in α-diversity between groups. Genus *Bacteroides* was associated with a lower risk of T2DM while genus *Sediminibacterium* increased the risk.	[26]
8.	Hypertension	Prospective cohort study (135)	Hypertension = 150; No hypertension = 150	16S rDNA (V6-V7 region) sequencing	At phylum level, phylum Proteobacteria was increased, and phyla Firmicutes and Bacteroidetes were reduced in hypertensive individuals. At genus level, genera *Acinetobacter* and *Sphingomonas* predicted a higher risk for hypertension. Genus *Staphylococcus* was associated with reduced risk of hypertension and its abundance positively correlated with HDL-cholesterol.	[27]

### 3.2. Cancers

Cancers are conventionally thought to be genetic diseases, however, recent studies have demonstrated unique intratumoral and blood microbiome profiles of certain cancer types (Table 2). Using whole genome and whole transcriptome data from The Cancer Genome Atlas (TCGA), a landmark study focused on sequencing reads that are not mapped to the human genome, and found that 35.2% of the unmapped reads (2.5% of the total reads) could be assigned to microorganisms such as bacteria, archaea and viruses [28]. These intratumoral microbial signatures were effective in discriminating normal tissue from tumors and different types of cancers, but did not correlate well with cancer stages in most cancer types [28]. More importantly, the blood microbial profiles of cancer patients were sufficiently sensitive to distinguish different cancer types, even for very early-stage cancers (stage Ia-IIc) and for tumors without notable mutational burden [28]. Data from independent studies also support the diagnostic and risk stratification capacity of circulating microbial profiles for hepatocellular carcinoma, myeloid cancers, and gastric cancer [29,30,31]. Taken together, the exciting results highlight a blood/intratumoral microbiome-based diagnostic method that may facilitate cancer screening and early detection.

Moreover, the bacterial genetic materials in the blood have been shown to predict the treatment response in cancer. In advanced colorectal cancer, the composition of baseline blood microbiota was significantly different between the immunochemotherapy (oxaliplatin + capecitabine + adoptive T cell immunotherapy) responders and non-responders with the genera *Bifidobacterium*, *Lactobacillus*, and *Enterococcus* more abundantly found among the responders [32]. The abundance of *Lactobacillus* was associated with longer overall survival [32]. Similarly, the enrichment of class Holophagae, family Peptostreptococcaceae, genera *Lewinella* and *Paludibaculum* predicted better clinical outcome and treatment response of nivolumab in advanced non-small cell lung carcinoma patients [33]. The data highlight the prognostic value of the circulating microbial profile in cancers.

The blood microbiome could act as a potential modifier of tumorigenesis and treatment effect. Specific β-glucuronidase and/or β-galactosidase microbes in the gut that can regulate estrogen metabolism, referred to as “estrobolome”, are increasingly known to increase the risk of estrogen receptor-positive breast cancer in postmenopausal women [34,35]. Similarly, dysbiosis of the estrobolome in the blood of breast cancer patients has also been reported. β-glucuronidase-producing bacteria were predominant in breast cancer patients, whereas β-galactosidase-producing bacteria were more abundant in healthy individuals [36]. Interestingly, *Staphylococcus,* which was depleted in the breast cancer group, was identified as a key modulator of anti-estrogen (i.e., tamoxifen) efficacy [36]. When estrogen-receptor positive breast tumor cells were cotreated with tamoxifen and *S. aureus*-derived extracellular vesicles, the AKT and ERK oncogenic signaling pathways were greatly suppressed, leading to more cell death [36]. Hence, variations in blood microbial diversity may indirectly regulate estrogen levels and breast cancer susceptibility. Further investigation is warranted to demonstrate the feasibility of blood microbiome engineering for the prevention and adjunctive intervention of breast cancer. In short, growing evidence supports the role of the blood microbiome in cancer diagnosis, prognosis, and therapy. Although the research is still in its infancy, these important findings may spur new advancements using blood-based microbial profiles for precision oncology. 

**Table 2 ijms-24-05633-t002:** Summary of investigations on the blood microbiome in malignancies.

No.	Disease	Study Design	Sample Size	Detection Method	Findings	Reference
1.	Pan-cancer	Re-analysis of TCGA whole genome (WGS) and transcriptomic (RNA-seq) sequencing data & cross-sectional patient cohort	TCGA: *n* = 10,481 (WGS; *n* = 4831 & RNA-seq; *n* = 13,285) In-house cohort: HIV- and cancer-free individuals = 69; Prostate cancer = 59; Lung cancer = 25; Melanoma = 16.	WGS, RNAseq and shotgun metagenomic sequencing	Cancer patients harboured significant microbial heterogeneity in the tumors and bloodstream that allowed the discrimination of different cancer types and between cancer and normal individuals, even at early cancer stages (Ia-IIc) or in tumors without detectable genomic alterations. Microbial profiles might not correlate to cancer stages. Plasma samples from independent cancer patient cohorts support the diagnostic capability of blood microbiome for cancers.	[28]
2.	Hepatocellular carcinoma (HCC)	Cross-sectional study	HCC = 158; Cirrhosis = 166; Healthy = 402	16S rDNA (V3-V4 region) sequencing	α-diversity was reduced in HCC patients. Genera *Staphylococcus, Acinetobacter, Klebsiella* and *Trabulsiella* were increased while *Pseudomonas, Streptococcus* and *Bifidobacterium* were reduced in HCC. A microbial signature characterized by five genera (*Pseudomonas, Streptococcus, Staphylococcus, Bifidobacterium,* and *Trabulsiella*) could discriminate HCC from healthy individuals with an area under the receiver-operating curve of 0.879, sensitivity of 0.729, specificity of 0.850 and accuracy of 0.816 when adjusted for age and sex.	[29]
3.	Gastric cancer	Case-control study	Gastric cancer = 71; Atypical hyperplasia = 6; Chronic gastritis = 11; Healthy = 13	16S rDNA (V1-V2 region) sequencing	Blood microbiome of gastric cancer patients were enriched by genera *Haemophilus, Bacteroides* and *Acinetobacter* with a depletion of *Sphingomonas, Comamonas* and *Pseudomonas*. *Alloiococcus*, *Salinicoccus, Pontibacter, Agrococcus* and *Enterococcus* were positively correlated with lymphatic metastasis and TNM staging of the tumors. Six microbial markers, including *Acinetobacter, Bacteroides, Sphingomonas, Comamonas, Haemophilus parainfluenzae,* and *Pseudomonas stutzeri*, resulted in an AUC of 0.858 with a sensitivity of 0.828 and specificity of 0.889 in a ROC analysis. Tumors with and without lymphatic metastasis showed differences in the abundance of *Haemophilus, Lautropia, Rosemonas, Enterococcus* and *Bacteroides*.	[30]
4.	Myeloid malignancies	Cross-sectional study	Acute myeloid leukemia (AML) = 612; Myelodysplastic syndromes (MDS) = 640; Myelodysplastic syndromes/myeloproliferative neoplasms (MDS/MPN) = 264; Myeloproliferative neoplasms (MPN) = 354; Healthy = 12.	WGS	Different myeloid malignancies demonstrated distinct blood microbiome profiles. Differences of microbial composition between AML and MDS are associated with host gene mutations and myeloblast cell percentages. The presence of Epstein-Barr virus is associated with poorer prognosis among low-risk MDS patients.	[31]
5.	Colorectal cancer	Retrospective cohort study	*n* = 39	16S rDNA (V3-V4 region) sequencing	α-diversity were higher for patients who responded to immunochemotherapy. At baseline, genera *Bifidobacterium, Lactobacillus,* and *Enterococcus* were more abundant, while *Pseudomonas* was diminished in responders compared to non-responders. *Lactobacillus* was associated with longer overall survival. Circulating bacterial DNA was negatively correlated with CD3-/CD16 + /CD56+ T cells in blood and decreased following immunochemotherapy.	[32]
6.	Non-small cell lung cancer (NSCLC)	Single-arm study	*n* = 72	16S rDNA (V3-V4 region) sequencing	Baseline abundance levels of class Holophagae, family Peptostreptococcaceae, genera *Lewinella* and *Paludibaculum* were associated with favourable clinical response to nivolumab in NSCLC patients. High Gemmatimonadaceae levels in the blood was linked to low response rate and disease progression.	[33]
7.	Breast cancer	Cross-sectional study	Healthy = 192; Breast cancer = 96	16S rDNA (V3-V4 region) sequencing	Breast cancer patients had higher abundance of genera *Citrobacter*, *Bacteriodes*, *Enterobacter* and *Bifidobacterium* while a depletion of *Staphylococcus, Lactobacillus, Fusobacterium, Porphyromonas* and *Actinomyces* in their blood stream. β-glucuronidase-producing bacteria were more abundant in the breast cancer patients. β-galactosidase-producing bacteria were predominant in the healthy individuals. Extracellular vesicles derived from *Staphylococcus aureus* improved the cytotoxicity of tamoxifen in estrogen-receptor positive breast cancer cells. AKT and ERK signaling pathways and cyclin E2 were suppressed while TNF-α was upregulated following adjunct therapy with *Staphyloccocus*-derived extracellular vesicles.	[36]

### 3.3. Liver Diseases

Most of the bacteria that translocate from the gut lumen into the blood circulation enter the liver via the portal vein system. Biologically, the liver acts as a sieve to trap and clear the invaders with the aid of local and systemic immune cells [37]. Hence, individuals with liver dysfunction, in particular fibrosis and cirrhosis, are more prone to septicaemia [37]. Recent studies have highlighted remarkable alterations in the blood microbiome in cirrhotic patients (Table 3). Liver fibrosis and cirrhosis markedly increase the abundance and diversity of circulating bacteria, many of which are gut commensals, suggesting a higher tendency of microbial translocation from the gastrointestinal tract [38,39,40,41]. For instance, the genus *Bacteroides* and family Enterobacteriaceae are highly enriched in cirrhotic patients compared to healthy individuals [39,40]. Interestingly, the microbiota of blood, ascites fluid and stool showed some commonalities between different anatomical sites, reinforcing the contribution of microbial translocation in shaping the blood microbiome in a diseased state.

Distinct blood microbiome profiles were detected at different venous compartments, namely, portal, hepatic, central and peripheral venous blood, in patients with liver cirrhosis (*n* = 7) who underwent a surgical intervention known as transjugular intrahepatic portosystemic shunt to restore the blood flow between portal and hepatic veins [42]. In comparison to peripheral blood, which had more genera *Brevundimonas* and *Mucilaginibacter*, there were more genera *Kocuria*, *Diaphorobacter* and *Paracoccus* found in hepatic blood, portal and central venous compartments, respectively [42]. Such a compartment-specific blood microbiome was not observed in another study [43]. The interstudy difference highlights possible interference from cirrhosis severity because at a more advanced stage, the hardening and scarring of the liver restricts blood flow and creates a unique microenvironment in the portal vein, which may favour specific bacteria. Surgical procedures such as shunting could potentially introduce microbes from other sources into the circulation, causing varying results to the blood microbiome profile. Hence, further experiments are needed to validate the concept of the compartment-specific blood microbiome. 

Certain circulating microbes are associated with the proinflammatory cytokine profile and nitric oxide signaling [39,42,43]. For example, the order Corynebacteriales is inversely associated with inflammatory cytokines, including IFN-γ, IL-17A and TNF-α [44]. A higher abundance of Corynebacteriales and a lower abundance of the genus *Massilia* may also predict the reversal of portal hypertension in hepatitis C virus (HCV)-induced cirrhosis after the completion of antiviral treatment [44]. These results point towards the role of certain microbes in the regulation of systemic and local vascular resistance through the stimulation of cytokine- and nitric oxide-mediated signaling cascades. In-depth investigation of the host–microbe interaction may reveal new strategies to mitigate inflammation and vasculopathy caused by progressive loss of liver function. 

**Table 3 ijms-24-05633-t003:** Summary of investigations on the blood microbiome in liver dysfunction.

No.	Disease	Study Design	Sample Size	Detection Method	Findings	Reference
1.	Liver fibrosis	Cross-sectional study (FLORINASH)	Spanish cohort: Liver fibrosis = 26; No fibrosis = 11 Italian cohort: Liver fibrosis = 11; No fibrosis = 60	16S rDNA (V1-V3 region) sequencing	16S rDNA concentration was higher in patients with liver fibrosis. Genera *Sphingomonas*, *Bosea*, and *Variovorax* were correlated with fibrosis severity.	[38]
2.	Cirrhosis	Case-control study	Case = 9; Control = 9	qPCR of conserved 16S rDNA region	Microbial diversity and DNA increased in cirrhotic patients. Gut commensals, including *Akkermansia muciniphila*, *Anaerostipes caccae*, *Bacteroides spp.*, *Enterococcus spp.* etc. were enriched in cirrhotic patients. Changes of blood microbiome upregulated genes for nitric oxide signaling.	[39]
3.	Cirrhosis with or without hepatocellular carcinoma	Case-control study	Case = 66; Control = 14	16S rDNA (V3-V4 region) sequencing	Cirrhotic patients had more *Enterobacteriaceae* and less *Akkermansia*, *Rikenellaceae* and *Erysipelotrichales*. Cirrhotic patients with hepatocellular carcinoma had more *Enterobacteriaceae* and *Bacteroides*, and less *Bifidobacterium*.	[40]
4.	Cirrhosis with or without ascites	Case-control study	Case (with ascites) =13; Case (without ascites) =14; Control = 17	16S rDNA (V4 region) sequencing	Blood microbial diversity was higher in cirrhotic patients with ascites. Order Clostridiales was more abundant while family Moraxellaceae was reduced in ascites patients. Ascitic fluid and fecal samples shared considerable bacterial taxa with blood microbiome.	[41]
5.	Decompensated liver cirrhosis receiving transjugular intrahepatic portosystemic shunt	Single-arm study	*n* = 7	16S rDNA sequencing	Portal, liver, central and peripheral venous blood showed compartment-specific microbiome. Inflammatory cytokines were associated with blood microbiome genera and formed patient-specific clusters.	[42]
6.	Cirrhosis with portal hypertension	Case-control study	Case = 58; Control = 46	16S rDNA (V1-V2 region) sequencing	Enrichment of *Comamonas*, *Cnuella*, *Dialister*, *Escherichia/Shigella*, and *Prevotella*, and reduction of *Bradyrhizobium*, *Curvibacter*, *Diaphorobacter*, *Pseudarcicella*, and *Pseudomonas* were seen in cirrhotic patients. Blood microbes from peripheral and hepatic veins were comparable. Abundance of *Escherichia/Shigella* and *Prevotella* correlated with IL-8 levels in the hepatic vein.	[43]
7.	HCV-induced portal hypertension	Single-arm study	*n* = 32	16S rDNA (V3-V4 region) sequencing	15 patients responded to antiviral treatment and had improved portal hypertension. Responders had more order Corynebacteriales and less genus *Massilia*. Corynebacteriales was inversely correlated with IFN-γ, IL-17A and TNF-α levels. *Massilia* was correlated with glycerol and lauric acid.	[44]

### 3.4. Respiratory Diseases

Studies on the blood microbiome in respiratory diseases are relatively scarce (Table 4). Two reports consistently found distinct bacterial compositions between asthmatic patients and healthy individuals [45,46]. The diversity of the blood microbiome in asthma patients demonstrated higher richness but reduced evenness [46]. Members of the phylum Bacteroidetes, namely *Bacteroides*, *Alistipes*, *Parabacteroides*, and *Prevotella*, were more commonly found in asthma patients, while the abundance of the phyla Actinobacter, Verrucomicrobia, and Cyanobacteria was slightly reduced [46]. When asthmatic patients were grouped by different inflammatory subtypes, including eosinophilic, neutrophilic, paucigranulocytic, and mixed granulocytic asthma, as well as by clinical features such as lung function and corticosteroid usage, varying bacteria were associated with each manifestation, highlighting the predictive value of blood microbes for patient stratification and personalized medicine. Indeed, differential blood bacteria profiles also enable accurate diagnosis of asthma up to a sensitivity of 0.94, specificity of 0.93 and accuracy of 0.92 [46]. Current data provide favourable prospects for the use of the blood microbiome as a diagnostic marker of asthma.

Unlike most studies that use 16S rDNA or shotgun metagenomics to study the blood microbiome, recent computational pipelines can infer microbial profiles from the RNA-sequencing of peripheral blood samples [47,48]. Using this approach, a microbial signature characterized by the genera *Acinetobacter*, *Serratia*, *Streptococcus* and *Bacillus* was found to be associated with increased dyspnea severity among former and current smokers [47]. Some of these enriched genera, i.e., *Acinetobacter* and *Streptococcus*, were independently reported in another study on tobacco smokers [49], thus providing insights into the reliability and accuracy of the inferred microbial abundance based on host high-throughput sequencing data. More importantly, the analytical approach readily enables the exploration of the host–microbe interactome through an integrative analysis of paired blood microbial profiles and host gene expression. Several genera, namely, *Streptococcus*, *Cutibacterium*, *Corynebacterium*, *Lactobacillus*, *Staphylococcus*, and *Bacillus*, were linked to biological activities such as oxidative phosphorylation and mTOR and WNT/β-catenin signaling in the circulating cells of the subjects [47]. Using a similar methodology, the inferred abundance of *E. coli*, *Bacillus sp.*, *Campylobacter hominis*, *Pseudomonas sp.*, *Thermoanaerobacter pseudethanolicus*, *Thermoanaerobacterium thermosaccharolyticum*, and *Staphylococcus epidermis* was correlated with the severity of COVID [48]. These bacteria also predict overactivity of the adaptive immune system and inflammatory response [48]. Taken together, the results not only shed light on the blood microbiome of smokers and COVID patients but also describe an innovative approach for the exploration of host–microbe interactions.

**Table 4 ijms-24-05633-t004:** Summary of investigations on the blood microbiome in respiratory diseases.

No.	Disease/Condition	Study Design	Sample Size	Detection Method	Findings	Reference
1.	Asthma	Case-control study	Case = 5; Control = 5	16S rDNA (V4 region) sequencing	Blood microbiome of healthy and asthmatic individuals was predominated by phyla Proteobacteria, Actinobacteria, Firmicutes, and Bacteroidetes. Asthma patients had altered microbiome communities more closely resembling oral and skin microbial profiles.	[45]
2.	Asthma	Case-control study	Case = 190; Control = 260	16S rDNA (V3-V4 region) sequencing	Asthma group had higher Chao 1, but lower Shannon and Simpson indices. 24 bacterial genera showed differential abundance between healthy and asthma individuals. *Escherichia/Shigella* was linked to higher eosinophilic asthma. *Comamonas* was linked to mixed granulocytic asthma. *Streptococcus* was linked to better lung function among the asthma patients. *Streptococcus, Rothia, Lactobacillus,* and *Staphylococcus* were associated with the usage of inhaled corticosteroid. *Bacteroides. Prevotella, Intestinibacter, Lactobacillus,* and *Blautia* were associated with the usage of systemic corticosteroid. A model using differential abundant blood microbes provided high sensitivity and specificity for asthma diagnosis.	[46]
3.	Smoking	Single-arm from a longitudinal cohort study (COPDGene study)	*n* = 3655 former and current smokers	RNAseq	Used Genome Analysis Toolkit (GATK) microbial pipeline PathSeq to identify microbial profile from RNA-seq of peripheral blood Proteobacteria, Actinobacteria, Firmicutes and Bacteroidetes were the major phyla in the blood. Inferred abundance of *Acinetobacter, Serratia*, *Streptococcus* and *Bacillus* were associated with dyspnoea scores. The analytical approach allows host–microbe interaction study.	[47]
4.	Smoking	Case-control study	Case = 20; Control = 21	16S rDNA (V4 region) sequencing	Blood microbiome in smokers had higher alpha-diversity index than in non-smokers. Many genera were enriched in smokers, including *Acinetobacter, Anaeroplasma, Bifidobacterium, Brachybacterium, Brevibacterium, Comamonas, Conexibacter, Fusobacterium, Geobacillus, Granulicatella, Lautropia, Macrococcus, Micrococcus, Porphyromonas, Rothia, Rubellimicrobium, Solobacterium, Streptococcus,* and *Veillonella* *Streptococcus* was enriched in both plasma and saliva of smokers.	[49]
5.	COVID-19	Meta-analysis of public transcriptomic data	17 PBMC of normal samples; 17 PBMC COVID-19 (GSE152418)	RNAseq	The inferred abundance of *E. coli, Bacillus sp., Campylobacter hominis, Pseudomonas sp., Thermoanaerobacter pseudethanolicus, Thermoanaerobacterium thermosaccharolyticum*, and *Staphylococcus epidermis* was positively correlated to the COVID severity and proinflammatory response. *Bacillus subtilis* was negatively correlated with COVID severity. *Bacillus subtilis* was linked to TCR signaling and humoral response.	[48]

### 3.5. Kidney Dysfunction

An aberrant blood microbiome could play a key role in kidney dysfunction, especially chronic kidney disease (CKD) (Table 5). A strong inverse correlation between glomerular filtration rate and the abundance of the circulating phylum Proteobacteria was observed [50]. CKD patients also exhibited a distinct blood microbial signature characterized by the enrichment of the genera *Legionella*, *Serratia*, *Yersinia*, *Acinetobacter*, *Pseudomonas*, *Lysobacter*, *Hyphomicrobium*, *Bacillus*, *Sediminibacterium* and *Pseudarcicella*, and the depletion of *Stenotrophomonas*, *Paracoccus*, *Sphingomonas*, *Tyzzerella*, *Corynebacterium* and *Candidatus* [50]. A comparison between CKD patients on peritoneal dialysis with and without vascular calcification highlighted the circulating level of the genus *Devosia* as a potential predictor of increased mortality risk [51]. In IgA nephropathy patients, blood *Legionella* and *Enhydrobacter* were highly abundant while *Staphylococcus* and *Streptococcus* were overrepresented when the eGFR ≤ 60 mL/min [52]. Collectively, the existing data reveal a possible implication of *Legionella* and *Devosia* in kidney impairment and nephropathy-related mortality, respectively. Interestingly, most of the circulating bacteria reported in patients with kidney dysfunction are not typical commensals of the urinary tract [53], implying that urinary mucosal disruption may not contribute to the dysbiotic blood microbiota. Hence, the origin and pathologic roles of disease-associated microbes should be examined to unravel their contribution to the onset of CKD and IgA nephropathy. 

### 3.6. Immune and Inflammatory Disorders

Dysbiosis in the blood may occur under inflammatory and immune diseases (Table 6), and these varying levels of bacterial populations may be used as biomarkers for such diseases. In systemic lupus erythematosus (SLE), an autoimmune disease, enriched levels of the genera *Desulfoconvexum, Desulfofrigus, Desulfovibrio, Draconibacterium, Planococcus* and *Psychrilyobacter*, and the phylum Gemmatimonadetes were identified [49,54]. Many of these were positively associated with the autoantibody levels in the plasma. Notably, exposure of heat-inactivated *Planococcus citreus* to peripheral blood mononuclear cells (PBMCs) resulted in heightened production of TNF-α, IL-1β, and IL-6 from monocytes, indicating that the enriched *Planococcus* in circulation may play a role in inflammation in SLE [49]. Likewise, distinct bacterial communities were found in the blood of rheumatoid arthritis patients, where the genera *Halomonas* and *Shewanella*, both implicated in inflammation and several human infections, were significantly elevated compared to healthy subjects [55]. Given that the perturbed circulating bacterial populations in both rheumatic diseases are primarily oral or gut commensals, the integrity of the mucosal barrier may be jeopardized, allowing more bacterial translocation and microbial-induced host inflammation [56]. Therefore, it is crucial to understand whether blood microbiome dysbiosis is a cause or effect of rheumatoid disorders.

Immunosuppression encourages the growth of opportunistic commensals, drastically changing the microbiota in the gastrointestinal tract and plasma [57,58]. In patients who received immunosuppressant regimens after liver transplantation, the levels of the families Anelloviridae, Nocardiaceae, and Microbacteriaceae increased, while that of Enterobacteriaceae decreased over a course of eight weeks post-operation [58]. The microbial profiles were sensitive to the use of antimicrobial agents. Changes in certain bacterial families, such as Xanthomonadaceae and Enterobacteriaceae, were associated with the occurrence of acute host–versus–graft rejection [58]. An ongoing prospective cohort study is currently looking into the role of the circulating microbiome in the pharmacodynamics and pharmacokinetics of various immunosuppressants (e.g., mycophenolate mofetil and tacrolimus) (NCT04953715), which may help to improve the success rate of organ transplantation in the future. 

HIV infection is another common factor of immunodeficiency. HIV carriers have higher 16S rDNA concentrations and diverse bacterial compositions in the blood than healthy subjects [49,59]. Many genera such as *Veillonella, Massilia, Haemophilus, Arthrobacter*, and *Fusobacterium*, were enriched in the blood of HIV patients, coupled with a reduction in *Altererythrobacter*, *Cryobacterium*, and *Anaerococcus* [49]. Coculture of peripheral blood mononuclear cells with either *Massilia timonae* or *Haemophilus parainfluenzae*, but not *Anaerococcus prevotii*, led to marked elevation of proinflammatory monocytes [49], highlighting that certain pathogenic bacteria may exacerbate the disease by driving chronic inflammation in HIV patients.

In terms of inflammatory diseases, Hyun et al. (2021) used a faeces-induced peritonitis porcine model and revealed the emergence of new circulating bacteria after disease induction, including *Escherichia/Shigella*, *Staphylococcus*, *Cloacibacterium*, *Diaphorobacter* and *Rhodanobacter* [60]. Functionally, these enriched bacteria are related to ABC transporters, oxidative phosphorylation, and two-component systems [60], which may aid in the pathogenesis of peritonitis. Interestingly, circulating levels of the genera *Escherichia/Shigella* and *Staphylococcus* were also significantly elevated in inflammatory bowel disease (IBD, comprising ulcerative colitis and Crohn’s disease) despite the unremarkable difference in the overall blood microbiome profiles between IBD and healthy individuals [61]. Hence, these common blood microbes could potentially contribute to the onset of inflammatory diseases in the abdominal cavity. 

Characteristic to the circulation of patients suffering from severe acute pancreatitis is the severe depletion of the phylum Actinobacteria and an abundance of the phylum Bacteroidetes compared to healthy subjects [62]. On a genus level, the changes translate to an increase in *Bacteroides*, *Stenotrophomonas*, *Serratia*, *Rhizobium*, *Prevotella*, *Staphylococcus*, and *Paracoccus*, with a stark depletion of *Acinetobacter*, *Lactococcus*, *Dietzia*, *Flavobacterium*, *Pseudomonas*, *Corynebacterium*, *Sphingobium*, and *Brevundimonas* [62]. Patients suffering from large vessel vasculitis (LVV) also show distinct blood microbiome compositions, with elevation of classes Cytophagia and Clostridia and unidentified taxa from the *Cytophagaceae* family with a suppression of genera *Zoogloeal* and *Staphylococcus* compared to healthy subjects [63]. The cytochrome P450 pathways were activated, while other biosynthesis pathways were suppressed [63]. Based on the data from independent studies, *Escherichia/Shigella* and *Staphylococcus* are consistently implicated in various inflammatory disorders, albeit with varying relative abundance levels compared to healthy subjects. These findings suggest a potential regulatory role of blood microbes in host inflammation which warrants further investigation.

**Table 6 ijms-24-05633-t006:** Summary of investigations on the blood microbiome in immune or inflammatory diseases.

No.	Disease/Condition	Study Design	Sample Size	Detection Method	Findings	Reference
1.	SLE	Case-control study	Case = 19; control = 30	16S rDNA (V4 region) sequencing	Genera *Corynebacterium*, *Desulfoconvexum*, *Desulfofrigus*, *Desulfovibrio*, *Draconibacterium*, *Ochrobactrum*, *Planococcus*, *Planomicrobium* and *Psychrilyobacter* were enriched in the circulation of SLE patients. Most of the enriched bacteria were positively correlated with plasma autoantibody levels. Heat-inactivated bacteria Planococcus increased TNF-α, IL-1β, and IL-6-producing monocytes in PBMC culture.	[49]
2.	SLE	Case-control study	Case = 11; control = 9	16S rDNA (V4 region) sequencing	SLE and healthy individuals demonstrated distinct β-diversity in plasma and gut microbial profiles. Phylum Gemmatimonadetes was enriched in the plasma of SLE patients.	[54]
3.	Rheumatoid arthritis	Case-control study	Case = 20; control = 4	16S rDNA (V4 region) sequencing	Genera *Halomonas* and *Shewanella* were more abundant while *Achromobacter*, *Escherichia-Shigella, Serratia, Corynebacterium 1, Streptococcus, Granulicatella, Gemella*, and *Staphylococcus* were reduced in the blood of rheumatoid arthritis patients. The circulating levels of *Haemophilus, Alloprevotella, Eremococcus*, and Lachnospiraceae_UCG001 were responsive to anti-rheumatoid drugs.	[55]
4.	Immunosuppression post liver transplant	Single -arm study	*n* = 51	WGS of cell-free DNA	Families Anelloviridae, Nocardiaceae and Microbacteriaceae increased while Enterobacteriaceae decreased gradually until 8-week post-transplantation. The use of antimicrobials affected the blood microbiome. Lower Enterobacteriaceae and higher Xanthomonadaceae were associated with acute cellular rejection.	[58]
5.	HIV infection	Case-control study	Case = 40; control = 51	16S rDNA (V4 region) sequencing	Genera *Veillonella, Massilia, Haemophilus, Arthrobacter*, and *Fusobacterium* were enriched in HIV individuals. Genera *Altererythrobacter, Cryobacterium*, and *Anaerococcus* were diminished in HIV patients. Coculture of *M. timonae* or *H. parainfluenzae*, but not *A. prevotii*, with PBMC increased the secretion of TNF-α, IL-1β, and IL-6 in monocytes.	[49]
6.	HIV infection	Two cross-sectional studies & one single arm study	227 HIV-infected patients; 15 healthy individuals	qPCR of conserved 16S rDNA region	Untreated HIV-infected patients had the highest copy number of 16s rDNA in the blood, followed by treated patients, and lastly healthy people. Bacterial 16S rDNA was associated to HIV viral load and circulating LPS in untreated patients. Higher 16S rDNA in HIV patients was linked to impaired CD4 T cell restoration.	[59]
7.	Peritonitis	Porcine experiment; pre- and post-fecal induced peritonitis	*n* = 6 domestic pigs	16S rDNA (V3-V4 region) sequencing	The profile of blood bacteria was relatively constant before and after peritonitis induction. Genera *Diaphorobacter, Rhodanobacter, Cloacibacterium, Escherichia/Shigella* and *Staphyococcus* increased gradually in the blood following peritonitis induction. The enriched bacteria were functionally linked to ABC transporters, two-component systems, and oxidative phosphorylation.	[60]
8.	Inflammatory bowel disease (including Crohn’s disease and ulcerative colitis)	Case-control study	Crohn’s disease = 8; ulcerative colitis = 8; control = 7	16S rDNA (V3-V4 region) sequencing	Bacterial DNA was isolated from extracellular vesicles in the blood stream. Microbial diversity and composition were comparable between healthy and diseased individuals. Genera *Escherichia/Shigella, Paracoccus, Xanthobacter, Pseudomonas, Lysobacter, Brachybacterium, Brevibacterium, Staphylococcus, Streptococcus, Lactococcus*, and *Lactobacillus* were elevated in some of the diseased patients, although further validation is necessary.	[61]
9.	Pancreatitis	Case-control study	Case = 50; control = 12	16S rDNA (V3 region) sequencing	Diversity richness was reduced in the blood microbiome of pancreatitis patients, characterized by increased Bacteroidetes and reduced Actinobacteria, which translated to an increase in Bacteroidia and Clostridia, and a reduction in Actinobacteriae, Flavobacteriia and Bacilli at the class level. Neutrophil-associated microbiome (based on microbial DNA extracted from neutrophils) were different between healthy and pancreatitis patients. Many neutrophil-associated microbes were associated to the abundance of T lymphocytes and reduction of neutrophilic bactericidal proteins in septic pancreatitis patients.	[62]
10.	Large vessel vasculitis (including giant cell arteritis & Takayasu’s arteritis)	Case-control study	Giant cell arteritis = 11; Takayasu’s arteritis = 20; Healthy = 15.	16S rDNA (V3-V4 region) sequencing	No difference in microbial diversity between healthy and arteritis individuals. Overall, arteritis patients had higher abundance of classes Cytophagia and Clostridia compared to healthy controls. Patients with Takayasu’s arteritis had more abundant classes Clostridia, Cytophagia and Deltaproteobacteria, and depleted Bacilli in the circulation. Patients with giant cell arteritis had more abundant genus *Rhodococcus* and family Cytophagaceae in the blood.	[63]

### 3.7. Pregnancy Complications

Linkages between the blood microbiome and pregnancy-related events such as preterm birth and stillbirth have been reported (Table 7). One study observed enrichment of the genera *Bacteroides*, *Lactobacillus*, *Sphingomonas*, *Fastidiosipila* and *Butyricicoccus*, along with a reduction in the abundance of *Delftia*, *Pseudomonas*, *Massilia* and *Stenotrophomonas* in mothers with preterm delivery [64]. Remarkable microbial differences were also found in the umbilical cord blood of preterm babies, as characterized by an increase in the genera *Fusobacterium*, *Actinomyces*, *Campylobacter*, *Peptostreptococcus*, *Porphyromonas* and *Prevotella* compared to term babies [65]. Furthermore, cord blood of stillbirths contained substantial amounts of bacterial 16S rDNA and heightened levels of *Streptococcus agalactiae/GBS* and *Arthrobacter* compared to live births [65]. The results point towards infection-induced intrauterine foetal death which is not widely recognized as a key driving factor of stillbirths [66]. 

Leveraging the concept of the multibiome, which delineates the interaction between different microbial entities (i.e., bacteria, fungi and viruses) as a community, a recent study revealed fascinating links between signatures of the blood microbiome and pathogens known to inflict pregnancy-related complications, namely *Toxoplasma gondii*, Hepatitis B virus, Human Papillomavirus (HPV), Rubella virus, Cytomegalovirus and Herpes simplex virus (HSV) [67]. For example, HPV positive blood samples were linked to elevated *Weissella paramesenteroides* and *Propionibacterium acne,* while blood samples positive for HSV had higher abundance of *Bifidobacterium bifidum* and *Propionibacterium acne* [67]. *Toxoplasma gondii* also has a high tendency to co-occur with *Pectobacterium carotovorum* and *Chthoniobacter flavus* in blood [67]. Essentially, the variations in the blood microbiome and co-occurrence of harmful pathogens with other microbes could be invaluable for the development of newer stratification strategy to identify mothers with a high risk for pregnancy-related complications via non-invasive maternal blood analysis.

**Table 7 ijms-24-05633-t007:** Summary of investigations on the blood microbiome in pregnancy complications.

No.	Disease/Condition	Study Design	Sample Size	Detection Method	Findings	Reference
1.	Preterm birth	Case control study	Case = 21; Control = 20	16S rDNA (V3-V4 region) sequencing	Phylum *Firmicutes* and *Bacteroidetes* were enriched while abundance of *Proteobacteria* was reduced in women with preterm delivery. Higher abundance of genera *Bacteroides, Lactobacillus, Sphingomonas, Fastidiosipila, Weissella* and *Butyricicoccus* in preterm birth blood samples.	[64]
2.	Stillbirth and preterm birth	Case control study	Stillbirth = 60; Preterm birth = 75 Live term (>37 weeks) birth = 101	16S rDNA (V4 region) sequencing	16S rDNA in umbilical cord blood was significantly higher in stillbirth and preterm birth Higher abundance of *Streptococcus agalactiae* and *Arthrobacter* detected in stillbirth Enrichment of *Fusobacterium nucleatum, Actinomyces, Campylobacter, Peptostreptococcus, Porphyromonas* and *Prevotella* in live early preterm birth Significant reduction in percentage of 13 ASV in live birth when gestational weeks increased from <32 to >37.	[65]
3.	Toxoplasma gondii, Others (Hepatitis B virus, Human Papillomavirus [HPV]) Rubella virus, Cytomegalovirus, Herpes simplex virus	Population-based cross-sectional study	*n* = 107,763 healthy controls	WGS of cell-free DNA	HPV positive contained significantly more *Weissella paramesenteroides* and *Propionibacterium acne* Herpes Simplex virus positive samples had increased enrichment of *Bifidobacterium bifidum* and *Propionibacterium acne* Toxoplasma gondii positive samples contained more *Pectobacterium carotovorum* and *Chthoniobacter flavus* No significant difference in blood microbiome of pregnant women having Rubella virus and Cytomegalovirus	[67]

### 3.8. Other Health Complications 

Blood microbiome dysbiosis has been detected in many other diseases (Table 8). Certain chronic dermatological disorders, namely, rosacea and psoriasis, are associated with distinct blood microbial profiles [68,69]. Enrichment of the genera *Staphylococcus*, *Sphingomonas*, and *Ralstonia* in psoriasis predicted the activation of lipid metabolic processes and adipocytokine signaling, which may contribute to chronic inflammation [69]. In contrast, patients with hidradenitis suppurativa, a chronic inflammatory skin complication characterized by recurrent nodules, abscesses, and scarring, carried blood microbiota that was similar to that of healthy individuals [70]. Hence, its pathogenesis could be unrelated to bacteraemia.

In patients with major depression, the genus *Janthinobacterium* was more abundant while *Neisseria* was reduced, but dysbiosis was ameliorated after antidepressant therapy [71]. A high abundance of Firmicutes, low abundance of *Tetrasphaera* and *Bosea* along with high tryptophan levels predicted a favourable treatment response to antidepressants [71]. The association of the blood microbiome with mood has also been discovered in patients with Parkinson’s disease. For instance, blood levels of the genera *Amaricoccus*, *Bosea*, *Janthinobacterium, Nesterenkonia* and *Sphingobacterium* were positively correlated with the Hamilton Anxiety Scale score, whereas *Aquabacterium*, *Bdellovibrio* and *Leucobacter* were positively correlated with the Hamilton Depression Scale score [72]. *Janthinobacterium* was also positively associated to the duration and disease severity score, along with other microbes such as *Cloacibacterium* and *Isoptericola* which were found to be enriched in patients with Parkinson’s disease [72]. Current evidence suggests a role of circulating *Janthinobacterium* in depression and anxiety. Hence, further studies are necessary to identify the cognitive-emotional implications of the bacteria. 

Additionally, a preliminary study isolated bacteria and fungi in their L-form (cell wall-deficient variant) from the blood of children with autism and their mothers [73]. The findings imply a vertical transmission of the pathogens from mother to child which may contribute to the onset of autism, but the results should be interpreted conservatively, as the proposed pathogenesis is largely speculation and statistical analysis was lacking in the study design. 

Disturbance of the blood microbiome has also been reported in polycystic ovary syndrome (PCOS), as demonstrated by the enrichment of families Nocardioidaceae and Oxalobacteraceae, and suppression of Burkholderiaceae, Lachnospiraceae, Bacteroidaceae, Ruminococcaceae and S24-7 [74]. 

Certain medical interventions may also escalate the risk of blood microbiome dysbiosis. For instance, surgical procedures are known to increase the risk of microbial invasion. Surgical patients who developed postoperative septic shock carried more abundant genera *Flavobacterium, Agrococcus, Polynucleobacter,* and *Acidovorax* in the blood, which were largely correlated with disease severity and organ failure assessment scores [75]. In contrast, total parenteral nutrition (TPN) administration confers profound shifts in the bacterial composition in the gut but less so in the blood [76]. In enterally fed mice, the expansion of the order Burkholderiales in the blood was likely due to catheter insertion [76]. 

**Table 8 ijms-24-05633-t008:** Summary of investigations on the blood microbiome in other health complications.

No.	Disease	Study Design	Sample Size	Detection Method	Findings	Reference
1.	Rosacea (skin disease)	Case-control study	Case = 10; control = 30	16S rDNA (V3-V4 region) sequencing	Rosacea patients had distinct blood microbial communities. Families Chromatiaceae and Fusobacteriaceae were elevated in rosacea patients. Genera *Rheinheimera*, *Sphingobium*, *Paracoccus*, *Marinobacter* were some of the most enriched blood microbes in rosacea subjects.	[68]
2.	Psoriasis (skin disease)	Case-control study	Case = 20; control = 8	16S rDNA (full length) sequencing	Psoriasis patients had lower bacterial diversity and richness and demonstrated a unique blood microbial signatures characterized by the enrichment of genera *Staphylococcus*, *Sphihgomonas*, and *Ralstonia*. The enriched blood microbes were linked to tryptophan metabolism, lipid biosynthesis, fatty acid metabolism, melanogenesis, and PPAR and adipokine signaling.	[69]
3.	Hidradenitis suppurativa (skin disorder)	Case-control study	Case = 27; control = 26	16S rDNA (V3-V4 region) sequencing	Patients with the skin disease had similar microbial diversity and composition compared to healthy individuals.	[70]
4.	Major depression	Case-control study	Case = 56; control = 56	16S rDNA (V3-V4 region) sequencing	Bacterial 16S rDNA concentration was comparable between depressive and healthy individuals. 13 genera, including *Actinomyces, Flavobacterium, Enterococcus, Neisseria, Tepidimonas, Aggregatibacter, Curvibacter, Fusobacterium* and a few unidentified genera were reduced in the patients. 5 genera including *Kocuria, Chryseobacterium, Parvimonas* and *Janthinobacterium*, were enriched in the patients. Abnormal abundance of *Neisseria* and *Janthinobacterium* were normalized following an antidepressant treatment. High levels of Firmicutes, low abundance of *Bosea* and *Tetrasphaera* and plasma tryptophan predicted favourable response to antidepressant. Bacterial activities linked to treatment response were related to xenobiotics, amino acids, and lipid and carbohydrate metabolism.	[71]
5.	Parkinson’s disease	Case-control study	Case = 103; Control = 104	16S rDNA (V3-V4 region) sequencing	The microbial composition and diversity richness were comparable between Parkinson’s disease and healthy individuals. Genera *Isoptericola, Cloacibacterium, Enhydrobacter* and *Microbacterium* were enriched while *Limnobacter* was reduced in Parkinson’s disease patients. Blood levels of *Amaricoccus, Bosea, Janthinobacterium, Nesterenkonia* and *Sphingobacterium* were positively correlated with Hamilton Anxiety Scale score while *Aquabacterium, Bdellovibrio* and *Leucobacter* were positively correlated with Hamilton Depression Scale score among Patkinson’s disease patients.	[72]
6.	Autism	Case-control study (without statistical analysis)	Case = 15 mother-child pairs; Control = 6 healthy individuals	Culture	Different species of L-form (cell wall-deficient) bacteria and fungi were isolated from autistic children and their mothers. L-form yeast and filamentous fungi could be reverted to their original form (with cell walls) under optimized culture conditions.	[73]
7.	Polycystic ovary syndrome	Case-control study	Case = 24; control = 24	16S rDNA (V3-V4 region) sequencing	Blood microbiome of PCOS women had reduced alpha diversity. Families Nocardioidaceae and Oxalobacteraceae were increased, while Burkholderiaceae, Lachnospiraceae, Bacteroidaceae, Ruminococcaceae, and S24-7 were decreased in PCOS group. Functional analysis predicted activation of metabolite transport proteins.	[74]
8.	Surgical-induced sepsis	Prospective cohort study	Healthy = 5; Non-infected = 7; Infected = 10; Sepsis = 18; Septic shock = 11	16S rDNA (V3 region) sequencing	Blood microbiome of infected, septic and septic shock patients were different from healthy individuals. Genera *Flavobacterium, Agrococcus, Polynucleobacter, Sphingomonas,* and *Curvibacter* were highly abundant in patients developing septic shock. Blood and neutrophil-specific microbiota in septic patients were originated from the gut. *Flavobacterium, Agrococcus, Polynucleobacter* and *Acidovorax* predicted deterioration of septicaemia.	[75]
	Total parenteral nutrition (TPN) administration	Mouse experiment	Chow-fed: 6; Chow-fed with jugular vein catheter insertion = 6; Intralipid-based TPN = 6; Omegaven-based TPN = 6	16S rDNA (V3-V4 region) sequencing	TPN induced remarkable gut microbial shifts, but fewer changes in the blood microbiome. Order Burkholderiales were enriched in the blood of mice with jugular vein catheter, which could be introduced during the insertion procedure.	[76]

## 4. Controversies and Counterclaims

While limited evidence supports the existence of a core healthy blood microbiome, dysbiosis of the blood microbial profile seems to have profound implications for a wide range of health conditions resulting from varying pathogenesis (Figure 2). The majority of the circulating microbes are associated with human body sites that are rich in commensals, such as the gut, oral cavity, and genitourinary tract [17,39,40,56,75]. Hence, the integrity and functionality of epithelial and mucosal barrier may be compromised in many diseases, allowing the opportunistic microbes or microbial-derived compounds to gain access into the body’s internal milieu [77]. Nevertheless, the concept of the blood microbiome remains a subject of debate as it challenges the paradigm of blood sterility. Here, we highlight the highly controversial and unresolved issues about the blood microbiome.

### 4.1. Susceptibility of Low-Biomass Samples to Exogenous Contamination

Many researchers remain sceptical about the presence of the blood microbiome because microbial analysis of low-biomass environments and samples are prone to contamination in most, if not all steps along the sample processing pipeline [78]. Several studies have concluded that circulating bacteria, especially those found in healthy volunteers, are either traces of microbial DNA from external sources or organic remnants from non-living bacteria [79,80]. For similar reasons, the findings of the microbiome in other low-biomass environments such as the womb during pregnancy, brain and tumors have also been brought into question [81,82]. Indeed, NGS sequencing is very sensitive to capturing minute amounts of microbial DNA contaminants from extraction kits, storage containers, sequencing reagents and even the sequencer itself. Moreover, dermal bacterial contamination during venepuncture and technical errors such as unintentional introduction of exogenous microbial genomes by nurses or lab personnel can also confound microbiome studies. Existing studies on the blood microbiome hardly report the decontamination strategy, and even when they do, the efforts may not be exhaustive because negative controls at every level ranging from sampling, extraction, library preparation to sequencing are likely needed. Augmentation of the decontamination strategy with bioinformatic and statistical approaches has been proposed to improve the chances of obtaining authentic microbial profiles [83,84]. Hence, developing a reliable and accurate methodology to address potential exogenous contamination serves as a cornerstone for microbiome research in low-biomass environments such as the bloodstream.

### 4.2. The Blood Microbiome: Viable Colonizers or Cell-Free DNA?

NGS offers a robust tool for culture-independent blood microbiome analysis; however, whether the resultant microbial profiles reflect the true blood colonizers or just the cell-free microbial DNA of killed pathogens from recent infection and spontaneous translocation remains debatable. Since the blood microbiome is often discussed in terms of its health implications, knowing its viability is indispensable to understand whether the microbiota has a larger role in pathogenesis or can merely serve as a biomarker. Nevertheless, very few studies have validated their microbiome results with the culture method, which is overly tedious and impractical because microbiome analysis easily yields hundreds of bacterial populations per sample, many of which could be fastidious and non-culturable [85,86]. Recent advancement in fluorescence in situ hybridization (FISH) on living bacteria enables validation of bacterial viability and taxonomic rank at the species level, although at much lower throughput compared to NGS [87]. Emerging bacterial single-cell transcriptomics such as PETRI-seq [88], microSPLiT [89], MATQ-seq [90] and BacDrop [91] are powerful tools that outperform 16S rDNA-based microbiota profiling and provide inferred viability and functionality derived from transcriptomes. The drawbacks are also evident: high cost and computationally demanding. Clearly, more efforts are warranted to establish whether the blood microbial profile represents living and active, or dead and non-functional microbes.

## 5. Knowledge Gaps and Future Directions

After reviewing the current evidence on the blood microbiome, we have identified several key gaps that may guide the future direction of the research topic. Firstly, there is a need to promote the integrative multibiome in the assessment of the blood microbiome. Current blood microbiome profiles are overrepresented by the bacteriome, while interest in the virome, archaeome and mycobiome is underwhelming. Compared to bacteria, the profiling of archaea, fungi and viruses is more challenging due to the paucity of analytical resources like reference genomes and established computational pipelines, thus posing substantial hurdles to those who are interested in the topic. However, the actual microbial ecosystem likely involves complex symbiotic relationships between microorganisms from different kingdoms that are well-implicated in the onset and pathogenesis of human diseases [92,93]. These exciting data strongly advocate the necessity to examine multibiome properties in the blood microbiome. Building fundamental infrastructures such as comprehensive genomic databases and analytical tools for the virome, archaeome and mycobiome is immensely beneficial to propel integrative multibiomics in microbiome research.

Next, our current understanding of the blood microbiome in human health is predominantly established in observational studies which only imply a correlation but not a causative relationship. Most of the studies have successfully identified specific groups of bacteria, the so-called core bacteria, that are enriched in different disease states. Subsequent studies should aim to decipher the disease-driving properties of these core bacteria and the underlying mechanisms. Germ-free animals and innovative in vitro 3D organoids are versatile models to characterize the host–pathogen interactions and explore the microbiome-based therapeutics [94,95].

Thirdly, there is an urgent need to establish the best practices and a standardized workflow for sample collection and processing, negative controls, and post-sequencing decontamination pipeline to enable data-sharing and inter-study comparison [81]. The gut microbiome is highly dynamic and labile to host and environmental factors such as host genetics, diet, medication, lifestyle and geographic regions [24,96]. Since blood microbes largely originate from the host commensals, it is not surprising that the blood microbiome is also affected by similar sets of factors, resulting in diverse core bacteria observed across studies on the same disease. By following a standardized workflow, data from independent studies could be pooled and meta-analysed with proper adjustment of the confounding variables. Alternatively, global-wide multi-center studies can also serve the same purpose, although they require international collaborative initiatives such as the International Human Microbiome Standards (IHMS) project. 

Lastly, the primary objective of blood microbiota profiling in different human diseases is to pinpoint their contribution to onset and progression. While the characterization of the potentially disease-driving core microbiome is informative, what is more critical is its collective biological functions and the impact on the host response. Currently, there are several packages for the prediction of metagenome functions, for example, PICRUSt [97], PICRUSt2 [98] and Tax4Fun2 [99]. However, significant biases have been detected in gene and functional inference, especially when the 16S rRNA-based metagenome is used as an input [100]. These tools also tend to predict general housekeeping functions but are less likely to reveal more specific adaptation pathways related to responses to environmental cues, secondary metabolites and xenobiotics, which are more biologically meaningful and disease-related [100]. Moreover, diverse core microbiomes may converge at the functional level and contribute to a common disease. This suggests a functional redundancy of microbes in different communities probably due to convergent evolution when exposed to similar environments. This phenomenon may add an extra layer of complexity to improve the performance of functional analytical tools. In short, newer functional prediction software should strive to combat the known biases and provide precise and biologically relevant functional estimation based on microbiome profiles.

## 6. Conclusions

To conclude, evidence supporting a core healthy blood microbiome is limited, yet dysbiosis of the blood microbial profile could have profound implications in a wide range of health conditions. Common microbial taxa have been identified in certain diseases, for example, *Legionella* and *Devosia* in kidney impairment, *Bacteroides* in cirrhosis, *Escherichia/Shigella* and *Staphylococcus* in inflammatory diseases, and *Janthinobacterium* in mood disorders. The blood microbial gene signatures also exhibit remarkable risk stratification and prognostication values in cancers, pregnancy-related complications, and asthma. The clinical implications of these findings require further validation as they are largely derived from observational studies.

## Figures and Tables

**Figure 1 ijms-24-05633-f001:**
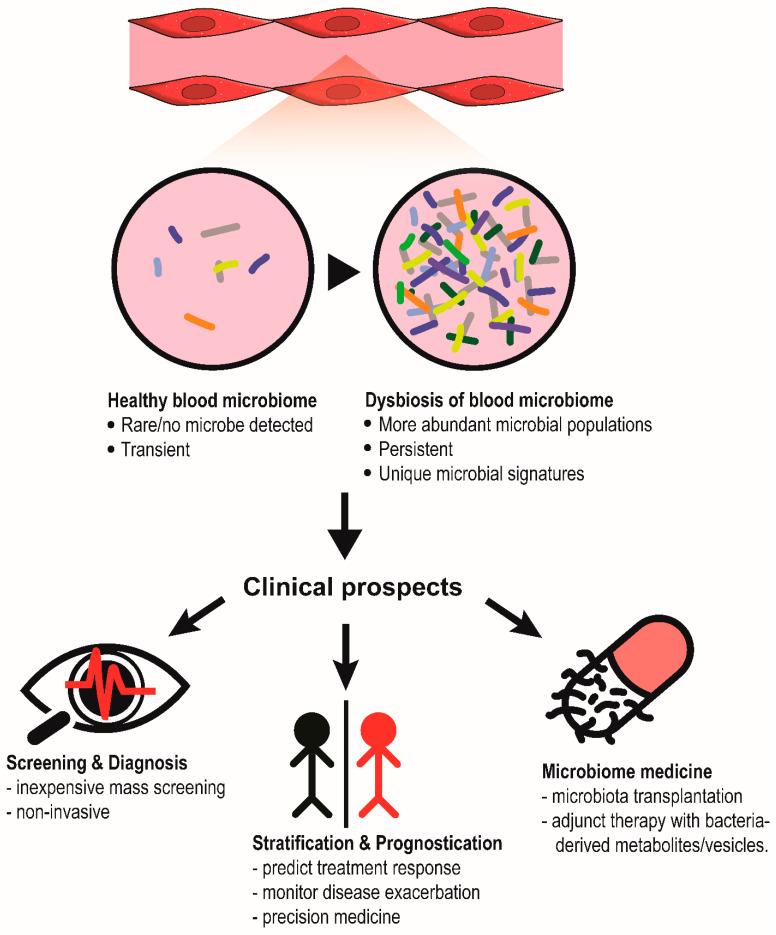
Clinical prospects of blood microbiome. In human diseases, the circulating microbial populations could become more abundant, forming a unique and persistent microbial signature that can be used for diagnosis, patient stratification or even the development of microbiome-based therapy.

**Figure 2 ijms-24-05633-f002:**
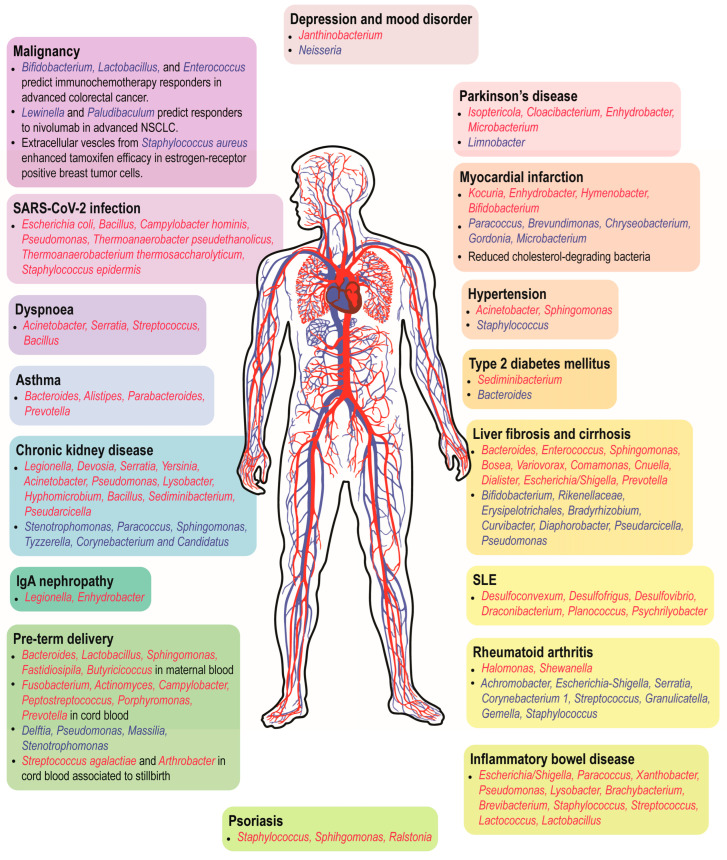
Circulating microbial populations at genus level that have been shown to associate with human diseases. Genera in red are enriched in patients with the diseases while those in blue are diminished/associated with beneficial effects in the patients. NSCLC, non-small cell lung carcinoma.

**Table 5 ijms-24-05633-t005:** Summary of investigations on the blood microbiome in kidney dysfunction.

No.	Disease	Study Design	Sample Size	Detection Method	Findings	Reference
1.	Chronic kidney disease (CKD)	Cross-sectional study	CKD = 20; Healthy = 20	16S rDNA (V3-V4 region) sequencing	16S rDNA concentration correlated with white blood cell count but showed no difference between CKD and healthy groups. Reduced α-diversity in CKD group Phylum *Proteobacteria*, class *Gammaproteobacteria* had significantly higher proportion in CKD group	[50]
2.	CKD patients on peritoneal dialysis (PD) with vascular calcification (VC)	Cross-sectional study	CKD-PD without VC = 12; CKD-PD with VC = 32	16S rDNA (V3-V4 region) sequencing	Blood of patients with VC had higher abundance of *Cutibacterium, Pajaroellobacter, Devosia, Hyphomicrobium.* Lower abundance of *Pelomonas* observed in blood of patients with VC. *Devosia* abundance in blood correlated to patients with high mortality risk.	[51]
3.	IgA nephropathy (Berger’s disease)	Case-control study	Case = 20; Controls = 20	16S rDNA (V3-V4 region) sequencing	Higher 16S rDNA levels in IgAN patients/ IgAN patients’ blood had more classes Coriobacteriia and Bacilli, genera *Legionella, Enhydrobacter, Staphylococcus* and *Streptococcus* compared to healthy controls.	[52]

## Data Availability

Not applicable.

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
