# Peer review of "The Blood Microbiome and Health: Current Evidence, Controversies, and Challenges"

_ijms, 2023, doi:10.3390/ijms24065633_

Round 1
Reviewer 1 Report
This review manuscript summarized current research in blood microbial profiles, pointing out correlations of blood microbiota and different diseases, providing potential biomarkers in diagnosis of health conditions. Moreover, authors described the challenges in studying blood microbiota, and expected future directions in this aera. In addition, authors claimed the controversies in existence, detection, and analysis of blood microbiota. The review was narrated clearly with a suitable structure. As blood is commonly thought to be sterile, blood microbiota in healthy controls may be unreliable, but blood microbiota in diseases can still be good references for disease prevention and treatment. One of the potential resources of blood microbiota comes from gut microbiota translocation due to gut leakage. So, it will be better if authors can discuss the linkage between gut microbiota and blood microbiota.
Author Response
Point-by-point response to reviewers’ comments
Reviewer 1
C1: This review manuscript summarized current research in blood microbial profiles, pointing out correlations of blood microbiota and different diseases, providing potential biomarkers in diagnosis of health conditions. Moreover, authors described the challenges in studying blood microbiota, and expected future directions in this aera. In addition, authors claimed the controversies in existence, detection, and analysis of blood microbiota. The review was narrated clearly with a suitable structure. As blood is commonly thought to be sterile, blood microbiota in healthy controls may be unreliable, but blood microbiota in diseases can still be good references for disease prevention and treatment. One of the potential resources of blood microbiota comes from gut microbiota translocation due to gut leakage. So, it will be better if authors can discuss the linkage between gut microbiota and blood microbiota.
R1: Thank you for the encouraging comments. We have added a few sentences (Lines 451-456) to discuss the possible origin of the blood microbes and its association with the commensals at other body sites. The underlying mechanism of the microbial translocation into the blood stream is a complex topic that have been summarized elsewhere [1]. Therefore, a relevant review on the translocation mechanism has been cited to provide further information on the topic.
Reviewer 2
C2: The article is well written, comprehensive, and provides valuable information regarding the role of microbiome in health and disease. The only critique I have is that Figure 2 needs to be redone. The print is not clear; the words indecipherable. Other than that, the review is an excellent contribution to the field.
R2: Thanks for the positive feedback. We have uploaded a new version of Figure 2 with a higher resolution and in tiff format.
Reference
- Akdis, C.A. Does the epithelial barrier hypothesis explain the increase in allergy, autoimmunity and other chronic conditions? Nat Rev Immunol 2021, 21, 739-751, doi:10.1038/s41577-021-00538-7.
Reviewer 2 Report
The article is well written, comprehensive, and provides valuable information regarding the role of microbiome in health and disease. The only critique I have is that Figure 2 needs to be redone. The print is not clear; the words indecipherable. Other than that, the review is an excellent contribution to the field.
Author Response

(The authors gave the same response as above.)
